# Debonding Detection of Reinforced Concrete (RC) Beam with Near-Surface Mounted (NSM) Pre-stressed Carbon Fiber Reinforced Polymer (CFRP) Plates Using Embedded Piezoceramic Smart Aggregates (SAs)

**Yang Liu [1,2], Ming Zhang [1] , Xinfeng Yin [1,\*], Zhou Huang [1] and Lei Wang [1]**

[1]  School of Civil Engineering, Changsha University of Science and Technology, Changsha 410114, China; liuyangbridge@163.com (Y.L.); zhangmingbridge@163.com (M.Z.); huangzhoubridge@163.com (Z.H.); leiwang@csust.edu.cn (L.W.)

[2]  School of Civil Engineering, Hunan University of Technology, Zhuzhou 412007, China

\*  Correspondence: yinxinfeng@163.com

**Abstract:** The application of reinforced concrete (RC) beam with near-surface mounted (NSM) pre-stressed carbon fiber reinforced polymer (CFRP) plates has been increasingly widespread in civil engineering. However, debonding failure occurs easily in the early loading stage because of the prestress change at the end of CFRP plate. Therefore, it is important to find reliable, convenient and economical technical means to closely monitor the secure bonding between CFRP and concrete. In this paper, an active sensing approach for generating and sensing stress wave by embedded smart aggregates (SAs) is proposed, which provides a guarantee for the secure connection between CFRP and concrete. Two specimens with different non-pre-stressed bond lengths were fabricated in the laboratory. Six SAs were installed at different positions of the structure to monitor the degree of debonding damage during the loading process. The experiments showed that the optimal length of non-pre-stressed CFRP bond section (300 mm) can significantly improve the load characteristics and enhance the service performance of the structure. The theoretical analysis of wavelet packet shows that increasing the length of non-pre-stressed CFRP bond section can slow down the occurrence and propagation of debonding cracks. The debonding crack in the tension end region is earlier than that in the bond end region. The research results reflect that the developed approach can monitor the damage process caused by debonding cracks and provide early warning for the initial damage and the debonding failure.

**Keywords:** carbon fiber reinforced polymer (CFRP); concrete; smart aggregates (SAs); active sensing approach; debonding detection

## 1. Introduction

The recent decade has seen the increasing applications of composite materials in civil engineering [1] and carbon fiber reinforced polymer (CFRP) has emerged as a novel alternative to conventional reinforcement in concrete structures because of its good fatigue endurance, light weight, high tensile strength and high corrosion resistance [2]. Therefore, reinforced concrete (RC) structure with near-surface mounted (NSM) pre-stressed CFRP plates have attracted more and more attention in civil engineering [3]. Firstly, CFRP plates can be easily pre-stressed and embedded in the pre-opened groove of the concrete surface [4]. Then the grooves are sealed with epoxy resin adhesives

to improve the overall bonding performance between the CFRP and the concrete, giving full play to the strength of CFRP material. This structure is a new form of application that has many advantages over conventional reinforcement methods. The CFRP material can be easily embedded in the concrete structure to improve the deformation resistance since three sides of concrete groove participate in the bonding. The negative moment area is easy to reinforce and the strength of the CFRP material is fully utilized to effectively improve the ultimate bearing capacity of the structure. In addition, the NSM method prevents the destruction of CFRP materials by fire, and, furthermore, the NSM process reduces the workload of concrete surface treatment.

Generally, CFRP strengthening technology includes external bonded method and NSM method [5]. Compared with the external bonded method, the advantages of NSM method have been proved [6]. The reinforcement technique of the NSM pre-stressed CFRP can solve the shortcomings of externally bonded CFRP and it is also proved to be one of the best reinforcement methods at present. But it is a difficult problem to find the best bond position of CFRP for reinforcement technology. At present, there are many studies on the debonding damage of CFRP-strengthened reinforced concrete structure but very few studies on RC structure with NSM pre-stressed CFRP plates have been reported. Therefore, this paper conducts an experimental research on the debonding damage of the RC beams with NSM pre-stressed CFRP plates. The optimal bonding position was found by using different CFRP bond section and health monitoring methods. However, the bearing capacity and safety of the structure are reduced due to overloaded vehicles, harsh environment, various corrosion [7], earthquake and aging of the structure. It is obvious that the demolition and reconstruction of these structures will cause great waste of resources, which is not suitable for the economic development of the country. Therefore, how to strengthen the structure reasonably has become a hot topic for experts and scholars in the civil engineering field. Moreover, it is an important subject in the field of civil engineering to use new materials to reconstruct and monitor the existing structure economically and reasonably.

However, the end of the CFRP plates is prone to stress concentration, which may lead to the appearance of debonding cracks in the early loading stage. Debonding failure initially occurs as very small cracks, which then propagate to other parts of the structure, ultimately leading to the brittle failure of the structure [8]. The accumulation of the following factors may cause brittle failure during extreme events: dynamic loads due to moving vehicles [9], corrosions [10] and impact loading [11]. It is difficult to take remedial measures in time without obvious warning before the failure, which may result in unwanted consequences. The brittle failure of the structure indicates that the CFRP plate is far below the ultimate tensile strength and the utilization ratio is low, which causes a great waste of material. Therefore, it is desirable to develop a reliable monitoring system to detect the interface bonding condition of RC structure with NSM pre-stressed CFRP plates.

The ultimate failure of the structure is caused by the debonding between the CFRP and concrete. Therefore, it is particularly important to monitor the initial debonding damage using an effective non-destructive testing (NDT) technique. In order to ensure a secure connection between CFRP and concrete, it is extremely urgent to find reliable, convenient and economical technical means to monitor the interface performance between CFRP and concrete. This can help to take further measures in time to avoid serious failure. At present, it is difficult to predict the failure of components under better linear behavior by using traditional strain and deflection measurement methods. Other NDT techniques in civil engineering include acoustic inspection technology [12], ultrasonic method [13], vibro-acoustic technique [14], infrared thermography [15], impact-echo approaches [16], microwave-based methods [17], ground penetrating radar techniques [18], fiber optic sensing [19] and among others. Di et al. have demonstrated the bearing mechanism and debonding damage process of FRP and self-compacting concrete by using acoustic inspection technology through experimental research [20]. Hsieh et al. discusses the feasibility of using the impact-echo method to assess the debonding flaws at the epoxy-concrete interfaces of near-surface mounted CFRPs [21]. However, this method requires the preparation of debonding defects for experimental research, which loses the randomness of structural damage. At the same time, the structural form commonly used in engineering is not adopted and only small-

sized specimen columns are prepared for testing. Kharkovsky et al. verified the feasibility of near-field microwave detection technology by monitoring the defects and debonding between CFRP and strengthened structures based on experimental studies [22]. These techniques are demonstrated to successfully monitor defects and debonding in the reinforced concrete structures strengthened with CFRP. However, the above-mentioned monitoring technology requires complicated algorithms, huge equipment and high cost, making online real time monitoring impossible.

The emergence of smart materials has the potential to overcome most of the shortcomings of traditional structural inspection techniques [23]. Piezoceramic materials are new smart materials widely used in recent years. Piezoceramic Lead Zirconate Titanate (PZT) has the advantages of wide frequency response range, fast response and low price. Based on the excellent characteristics, piezoceramic PZT transducers are not only widely used in communications, aerospace, radar, ultrasound and other fields but also provide a new research direction for civil engineering health monitoring. The price of structural health monitoring (SHM) based on piezoceramic PZT is low and extensive application in the field of civil engineering can bring good economic benefits. Therefore, many scholars have conducted extensive research on the health monitoring of CFRP-reinforced structures based on piezoceramic transducers. Furthermore, piezoceramic materials can also enable electromechanical impedance (EMI) technology [24] and the smart aggregates (SAs) based active sensing method in SHM [25]. By comparing the changes of electrical impedance signals before and after structural damage, EMI technology can be used to diagnose the structural damage due to the electromechanical coupling characteristics of piezoceramics [26]. Active sensing technology judges the damage condition of the structure according to the change of the signal in the transmission process. The SAs based active sensing methods have many advantages such as sensitivity to the initial damage of the effects of sensing and driving. Thus, compared with EMI technology, the algorithm is easy to understand and analyze and does not need to use large-scale equipment. Xu et al. Proposed an SA-based active sensing method for monitoring the bonding performance of GFRP bars and concrete structures through experiment research [27]. Wang et al. prepared different types of damage in CFRP-reinforced concrete columns, which proved the feasibility of monitoring the CFRP-concrete interfacial damage based on the active sensing method of PZT patch and SA [28]. However, the research on the method and mechanism of monitoring the interface damage of CFRP-concrete by piezoceramic transducer is still limited. In addition, most studies only observed the beginning of debonding; how the damage developed to induce debonding and the expansion of the debonding region have not yet been realized. So far, the research methods for the location and severity of debonding damage have not formed a complete set of system. Although researchers found that the self-sensing of the piezoceramic transducer can effectively monitor the CFRP-concrete interfacial damage, the damage type is difficult to identify and the feasibility of damage detecting needs further verification in practice.

Anchoring of the end zones has been one of the biggest problems in the RC beams with NSM pre-stressed CFRP plates. To prevent debonding failures, the CFRP must be mechanically anchored. Different techniques using bolted metal plates have been tested as well as decreasing the thickness of the CFRP at the ends to lower the stresses in the concrete-CFRP interface. The results have proven effective. Multilayer application of CFRP has been tested to achieve a different prestressing profile on the concrete beam [5,6]. Peng et al. determined the optimal location of CFRP bond based on a large number of experimental studies [5,6]. However, the location of FRP jackets is very important factor towards a resilient design. Mahdavi et al. proposed an optimized strengthening scheme for reinforced concrete frame structures using plastic hinges of CFRP constrained columns with different layers [29]. A single objective function was defined and the optimization was carried out by two different meta-heuristic algorithms-namely, genetic algorithm and particle swarm optimization. This technique enhances their ductility and increases the resiliency of the structure. In the previous research and analysis, the strengthening method of CFRP is determined and the influence of the applied prestress, the size and quantity of CFRP and the anchoring method on the strengthening structure is discussed. In most experiments, the bond length of CFRP is the same as that of the structure, so

the engineering application and economic problems are ignored. The optimal arrangement of CFRP needs to be solved. Predetermined CFRP application modes are used in most CFRP reinforcement evaluation researches. Few researches involve optimization methods based on structural properties and thus propose optimized retrofit solutions to save the cost of CFRP materials. Most of the research on CFRP reinforcement has been conducted on independent components, rather than treating them as part or whole of the structure. However, in practical engineering, engineers are most concerned about the response of the structure as a whole. Therefore, the performance evaluation of the overall structure before and after CFRP strengthening is a very demanding research field. In this paper, the optimal position is found by different bonding length and health monitoring method based on SA. The number of CFRP required for the structure should be minimized and the reinforcement requirements should be met.

To the best knowledge of the authors, the debonding monitoring of RC beam with NSM pre-stressed CFRP plates using piezoceramic transducer enabled active sensing has not been reported. In this paper, we explore the research of active sensing based debonding monitoring of reinforced concrete beam with near-surface mounted pre-stressed CFRP plates using embedded piezoceramic smart aggregates. To enable the experimental study, two RC beams with NSM pre-stressed CFRP plates were fabricated. The tensile region under the beams can be divided into pre-stressed CFRP bonded section, non-pre-stressed CFRP bonded section and unbonded section. Load tests were carried out on the specimens embedded with SAs for monitoring crack development and interface debonding damage. Different lengths of the non-pre-stressed CFRP bonded section were used as experimental parameters in the experiment. The load characteristics and stiffness of the test specimens were measured by vertical displacement and strain during the loading process. The occurrence and severity of cracks weaken the propagation energy of the stress wave. The active sensing approach was used to monitor the crack development and interface debonding damage of the specimens under vertical loading. The time domain, frequency domain analysis and further wavelet packet analysis were investigated. At the same time, the feasibility and superiority of the approach were verified by comparing with the deflection change and strain data of the beams.

## 2. Monitoring Principle

### 2.1. Piezoceramic Smart Aggregates

The most typical feature based on piezoelectric materials is the piezoelectric effect, which has been widely used in SHM. PZT has become a popular piezoelectric material due to its strong piezoelectric effect. Due to the special piezoelectric properties, PZT with dual functions of stress wave transmission and detection can be utilized as both actuator and sensor [30]. Since the PZT is fragile, special treatment methods are usually adopted to protect the PZT from working properly. In the experimental study in this paper, the type of PZT-5A piezoceramic patch with both electrodes on the same side is required to be in the compressive vibration mode because the signal propagates in the structure in the form of longitudinal waveform. The size of PZT patch is 15 mm × 10 mm × 0.3 mm, as shown in Figure 1.

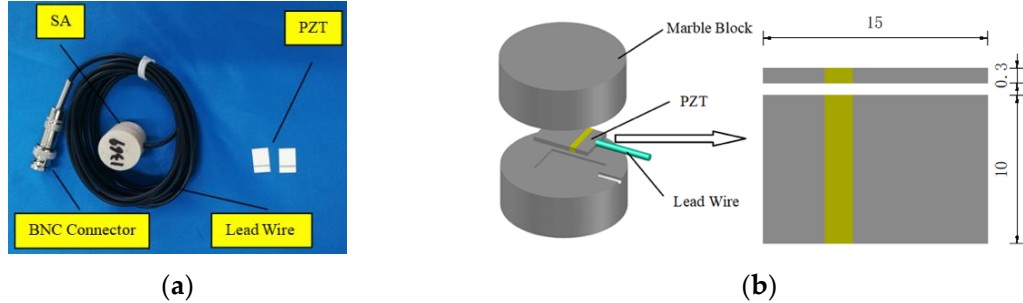

(**a**)　　　　　　　　　　　　　　　　　　　　　　(**b**)

**Figure 1.** (**a**) A photo of a smart aggregate (SA) and Piezoceramic Lead Zirconate Titanate (PZT) patch; (**b**) the schematic structure of the SA.

The smart aggregate (SA) was made by sandwiching PZT patch with lead wires between two marble blocks using epoxy resin [27]. For the connection, one end of the lead wire is soldered to the PZT patch and the other end is connected to the Bayonet Neill–Concelman (BNC) connector. The data can be easily transmitted to the instrument through the BNC connector. The fabricated piezoelectric sensor is shown in Figure 1. The diameter of the SA is 25 mm and the height is 20 mm.

### 2.2. Wavelet Packet-Based Active Sensing Method

The signal analysis in the structural damage identification technology using the active sensing analysis method mainly includes time domain analysis, frequency domain analysis and wavelet packet analysis [28]. The time domain analysis mainly investigates the change of signal amplitude to identify structural damage. Time domain waveform is intuitive and easy to understand, so time domain analysis is widely applied in engineering. Structural damage will cause the change of signal frequency. The damage can be analyzed according to the change of signal frequency spectrum. The frequency domain signal is more intuitive than the time domain signal in the change of the signal amplitude. Therefore, we transform the time domain signal data into frequency domain signal by Fourier and perform comparative analysis [31]. Wavelet packet analysis can identify damage by comparing the energy difference of the received signal. In recent years, wavelet packet analysis has proven to be an effective SHM method. Wavelet packet analysis has the advantages of time domain and frequency domain analysis and can accurately decompose the low frequency and high frequency. The special characterization of the signal based on wavelet packet analysis can quantitatively describe the severity of structural damage during the loading process. Therefore, this paper can analyze the signal by wavelet packet principle and form the damage index to identify the severity of structural damage.

The energy of loading process is calculated based on wavelet packet theory and the damage index is determined by root mean square deviation (RMSD) to evaluate the damage extent of the specimen. In order to eliminate the initial error of different sensors, the damage index is normalized. The energy loss of stress wave propagation is caused by structural damage (such as cracking, debonding and bond slip) at different loading stages. The structure is in the state of no debonding and cracking before the loading test, indicating that the damage index value is zero. The damage index reaches a certain value, indicating that the structure will exhibit debonding/cracking during the loading process. When the damage index is closer to 1, the structural state indicates that the damage extent is serious.

The wave components are generated by the wavelet packet decomposition of the wave signals obtained from the experimental study. The "Daubechies (db2)" wavelet packet provided by Matlab is selected in the process of wavelet packet analysis. The original signal measured by SA is decomposed into three layers of wavelet packets. Three level decomposition diagram of wavelet packet, as shown in Figure 2.

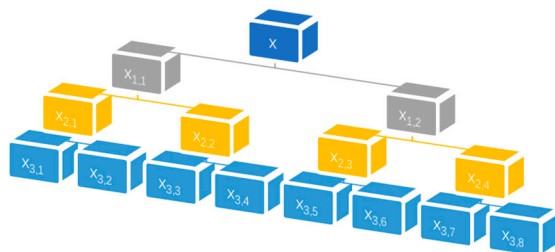

**Figure 2.** Three level decomposition diagram of wavelet packet.

## 3. Experimental Study

### 3.1. Specimen Details and Fabrication

In order to monitor the cracking characteristics of RC beams with NSM pre-stressed CFRP plates, two test specimens were designed with the same dimensions and reinforcement. The total length of each

beam is 2700 mm and the cross section is 150 mm × 250 mm. There are two CFRP plates with a spacing of 80 mm at the bottom of the beam, which are bonded by three sides of epoxy resin. The thickness of top and bottom protective layer is 50 mm and 30 mm respectively. The tensile area of the beam is divided into pre-stressed CFRP bonded section, non-pre-stressed CFRP bonded section and unbonded section. The boundary between the pre-stressed CFRP bonded section and the non-pre-stressed CFRP bonded section is called the tensioned end and the junction of the non-pre-stressed CFRP bonded section and the unbonded section is called the bonded end. The total bond length of CFRP plate in beam 1 is 1800 mm, the length of pre-stressed bond section is 1400 mm and the length of non-pre-stressed bond section on both sides of beam is 200 mm, as shown in Figures 3 and 4. The total bond length of CFRP plate in beam 2 is 2200 mm, the length of pre-stressed bond section is 1400 mm and the length of non-pre-stressed bond section on both sides of beam is 400 mm, as shown in Figures 4 and 5. The test specimens in the experimental program were made of concrete, CFRP plate, epoxy resin, longitudinal distributed reinforcement, steel stirrups and SAs. Strain gauges were symmetrically arranged on the outside of the CFRP plate at the non-pre-stressed CFRP bond section of the beam to measure the strain of the CFRP plate during the loading process, as shown in Figures 3 and 5. The cement used for casting the specimen is type 32.5 Portland cement [30]. The mixture ratio of the concrete is shown in Table 1. The average compressive strength of the concrete for 28 days is 30 MPa. The thickness and width of CFRP plate are 2 mm and 16 mm respectively. The initial prestress applied was 900 MPa. The mechanical properties of CFRP and epoxy resin are listed in Table 2. In the research, the steel stirrups are arranged with a spacing of 100 mm along the length of the test specimen. Table 3 shows the mechanical properties of the steel. The SA is installed in the tension zone of the beam, as shown in Figures 3 and 5.

Two beams with different lengths of non-pre-stressed CFRP sections were fabricated and there were pre-stressed and non-pre-stressed sections. Therefore, the experimental procedure involves two different phases of comparative experimentation: (1) Monitoring the cracking characteristics; (2) Monitoring the crack development characteristics. The entire specimen is divided into four observation surfaces (OS), as shown in Figures 3 and 5. The signal received by the SAs is compared with the concrete cracking and crack development and failure at the bonded end region and tensioned end region of the beam.

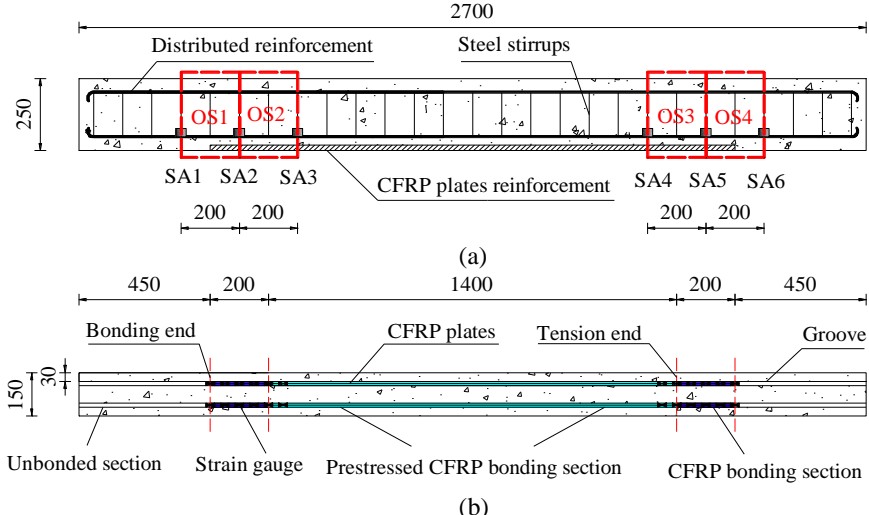

**Figure 3.** Test beam 1 details (unit: mm): (**a**) Side view (**b**) Horizontal view.

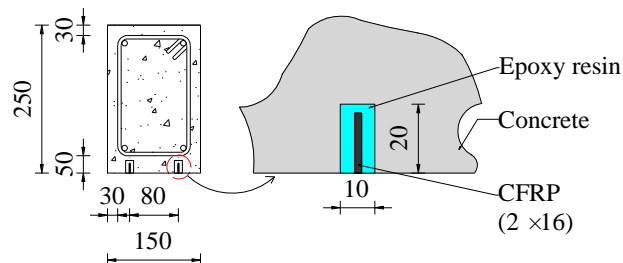

**Figure 4.** Cross section details (unit: mm).

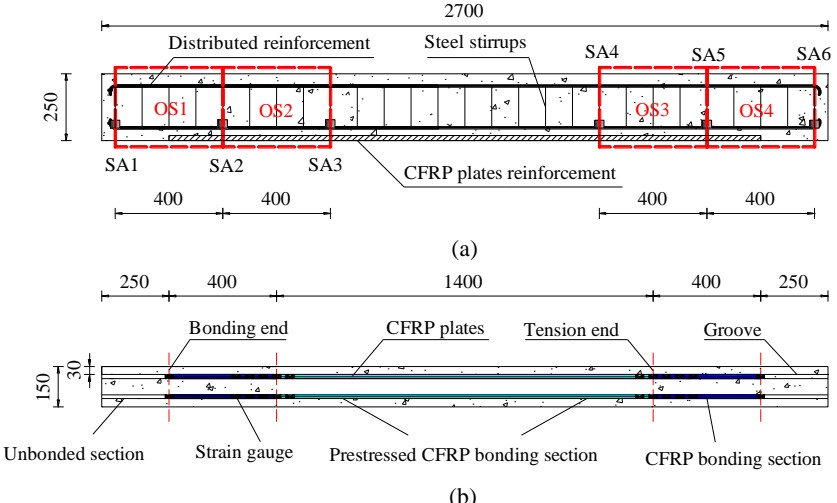

**Figure 5.** Test beam 2 details (unit: mm): (**a**) Side view (**b**) Horizontal view.

**Table 1.** Mixture ratio of the concrete.

| Cement (kg/m$^3$) | Water (kg/m$^3$) | Sand (kg/m$^3$) | Stone (kg/m$^3$) | Water Reducing Agent (kg/m$^3$) |
|---|---|---|---|---|
| 372 | 175 | 815 | 996 | 7.4 |

**Table 2.** Mechanical properties of CFRP and epoxy resin.

| Material Type | Tensile Strength (MPa) | Tensile Modulus (GPa) | Shear Strength (MPa) |
|---|---|---|---|
| CFRP plate | 2068 | 140 | - |
| Epoxy resin | 24~27 | 11.2 | 14~17 |

Note: the epoxy resin material can achieve the above properties in 7 days at + 15 °C.

**Table 3.** Mechanical properties of steel.

| Type | Diameter (mm) | Yield Strength (MPa) | Ultimate Tensile Strength (MPa) | Elastic Modulus (GPa) |
|---|---|---|---|---|
| Longitudinal distribution reinforcements | 16 | 400 | 540 | 201.9 |
| Steel stirrups | 8 | 335 | 445 | 200 |

### 3.2. Instrumental Setup

The experimental equipment for CFRP-concrete interface debonding monitoring used in this paper mainly include RC beams with NSM pre-stressed CFRP plates, SAs, reaction frame, load cell, screw jack, steel pad, load distribution beam, concrete support piers, fixed support, sliding support, dial indicators, multifunctional strain gauge, data acquisition system (NI-USB 6366), laptop with support software, as shown in Figures 6 and 7. The strain change of the CFRP plates during the loading

process was measured using a multifunctional strain gauge. The deflection was measured by dial indicator, the load was applied to the beam with a screw jack and the load value was measured with a load cell.

The NI-USB 6366 generates a swept sine wave signal to excite the actuator SA, while the sensor SA records the wave signal through a data acquisition system with a sampling frequency of 1 MHz. The designed swept sine wave signal has a frequency range of 100 Hz to 150 kHz. The amplitude and period of the swept sine wave signal of the excitation actuator are 10 V and 1 s, respectively. NI-USB 6366 data acquisition system is supported by NI LabVIEW software. Data acquisition and storage are implemented by NI LABVIEW software. Matlab software is used for data analysis. The signal filtering is processed by a bandpass filter with a cutoff frequency of 50,000 Hz to 150,000 Hz.

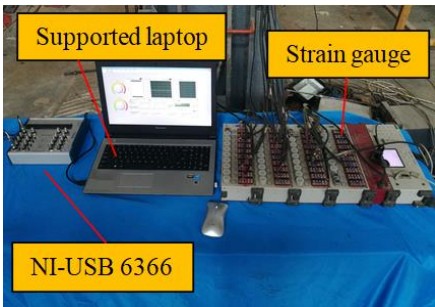

**Figure 6.** Experimental setup.

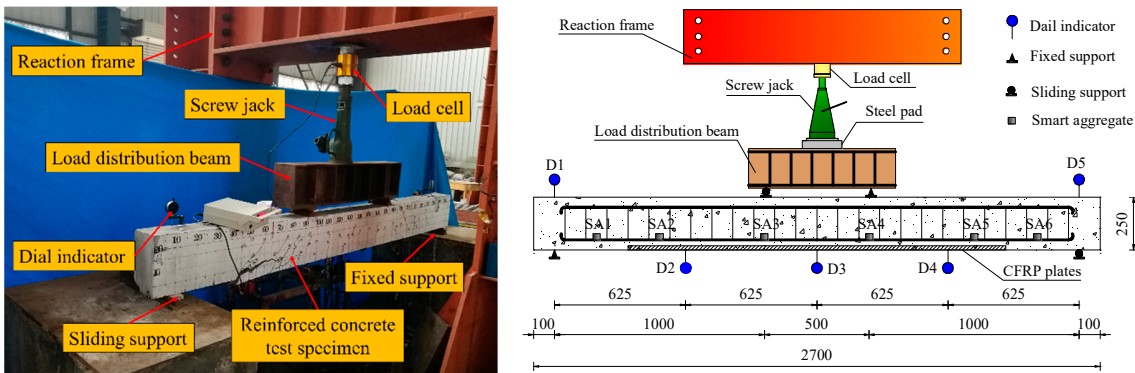

**Figure 7.** Specimen Loading System.

### 3.3. Experimental Procedures

This experiment adopts the loading method of monotonic loading static test. The load generated by the screw jack through the fixed reaction frame is transmitted to the specimen through the steel pad, load distribution beam and support. The loading is performed by a two-point loading method and the concentrated load is converted into two equal-sized forces by the distribution beam. Detailed layout of the experimental loading is shown in Figure 7. Before the formal loading test, in order to eliminate the initial defect of the sample and the normal operation of the instrument, the preloading test is carried out. The formal loading of specimen starts from 0 kN until complete failure. The increment of each load level in the loading process is 5 kN to observe the occurrence and development of cracks in the specimen. The data was collected in increments of 10 kN for each load level in the test. Before loading, the actuator is excited by the sweep signal and the data in healthy state was collected by the sensor. During the loading process, data are collected once for each load level and multiple measurements are taken at the same location. Meanwhile, strain data were collected and deflection changes were recorded. After each stage of loading, the specimen would have sufficient deformation time and the data would be measured after deformation stability.

## 4. Experimental Results

### 4.1. Load Characteristics of the Specimens

The photos of the failed beam 1 and beam 2 are shown in Figures 8 and 9. The number of cracks in beam 2 is more than that in beam 1 during the test loading process. With the increase of CFRP reinforcement, the crack spacing decreases and the number of cracks increases. By comparing the failure photos of the test specimens, it can be seen that the debonding failure degree of the beam 2 is more severe than that of the beam 1 and the cracks only appear in the CFRP reinforcement area during the loading process. This is mainly because the non-pre-stressed CFRP bonded section of the beam 2 is twice that of the beam 1. This indicates that the increase in the contact area between the CFRP and the concrete improves the stress distribution of the concrete, slows down the appearance and expansion of the crack and improves the service performance of the test specimens. When the same load is applied, the beam 1 has a larger displacement and a more severe crack than the beam 2. Load characteristics of the specimens are shown in Table 4. It can be seen from Table 4 that the cracking load and ultimate load of beam 2 are significantly higher than those of beam 1. Compared with beam 1, the cracking load of beam 2 at the tension and bond end is increased by 50% and 177.78% respectively and the ultimate load is increased by 23.81%. Beam 2 exhibits better ductile failure characteristics due to the increased length of the non-pre-stressed bond section. The analysis shows that the load characteristics of the specimens can be improved by increasing the length of the non-pre-stressed CFRP bonded section. Figure 10 shows the load–vertical displacement curve in the mid-span of the specimen. There is little difference between the load–vertical displacement curves of beam 1 and beam 2 at the initial stage of loading. The applied load–vertical displacement curves appear an obvious change at the cracking load. The vertical displacement of beam 2 is significantly smaller than that of beam 1 under the same applied load. The ultimate vertical displacement of beam 1 and beam 2 during debonding failure is 14.88 mm and 22.53 mm, respectively. This indicates that the stiffness of the specimen is obviously improved after increasing the length of the non-pre-stressed CFRP bonded section. The non-pre-stressed bonding section can prolong the bond stress-transmission length of the CFRP-concrete interface and can significantly enhance the bearing capacity and deformation performance of the test specimen. The CFRP axial strain distribution curves of beam 1 and beam 2 are shown in Figure 11. It can be seen from Figure 11 that the strain peak in the non-pre-stressed CFRP bonding section exhibits a tendency to move from the tension end to the bonding end with the increase of load. The strain of beam 1 bonded end is about 400 $\mu\varepsilon$ at the 100 kN load stage, indicating that the strain is sufficiently transmitted within the 200 mm length of non-pre-stressed CFRP bond section. The failure photo of the non-pre-stressed CFRP bond section of beam 1 in Figure 8 also proves this phenomenon. The strain difference of beam 2 does not change from 110 kN to the failure load stage, indicating that the non-pre-stressed CFRP bond section does not function to transmit the interface bond stress caused by the tension and the load. Strain analysis showed that there was an effective non-pre-stressed CFRP bond length to improve the load bearing properties of the specimen. The released tensile prestress is equivalent to an axial drawing force for the non-pre-stressed bonded section, so the performance of the test specimen increases with the length of the non-pre-stressed bonded section. According to the experimental observation, the crack only appears within the length of 300 mm of the CFRP non-pre-stressed bonding section of the beam 2 in the 0 kN to 110 kN loading phase. The bonded end of the beam 2 is cracked during the 125 kN loading phase. The beam 2 undergoes debonding failure from the non-pre-stressed CFRP bond end to the mid-span direction during the loading stage of 130 kN. The failure photo of beam 2 in Figure 9 proves this phenomenon. The results show that the bearing capacity and deformation performance of the structure increases with the extension of the non-pre-stressed CFRP bonding section and the performance improvement of the structure is limited when the effective bond length (300 mm) is exceeded. From the analysis of deflection and strain, it is impossible to make an effective prediction of the degree of debonding damage.

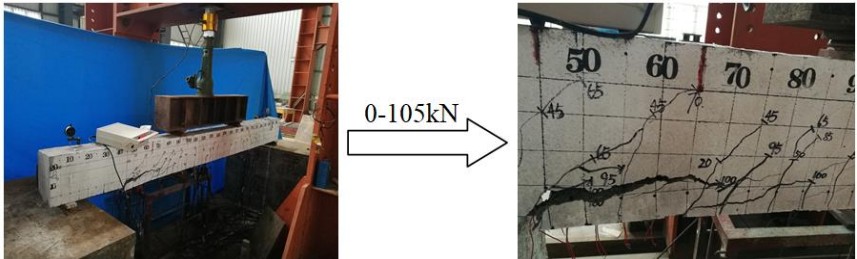

**Figure 8.** Failure photos of the beam 1.

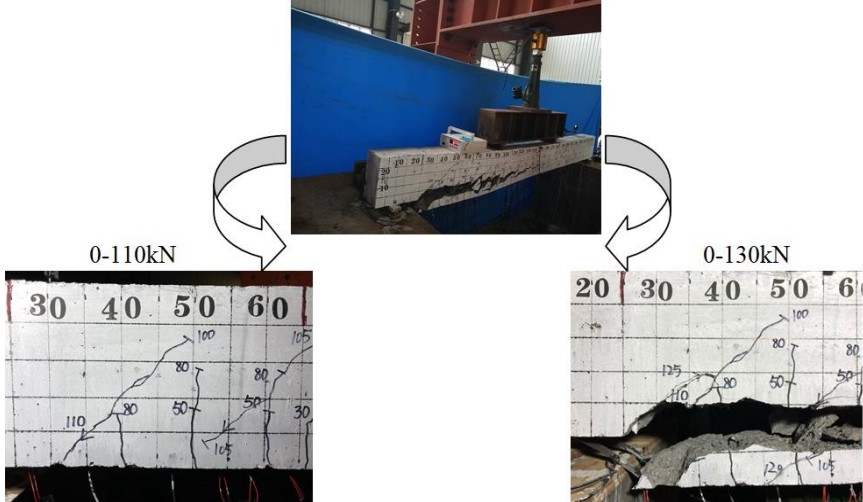

**Figure 9.** Failure photos of the beam 2.

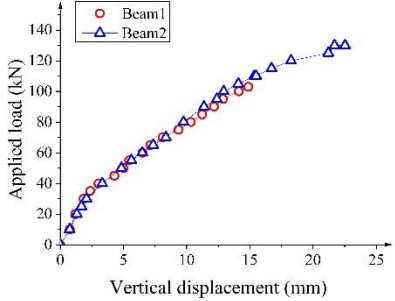

**Figure 10.** The applied load–vertical displacement curves.

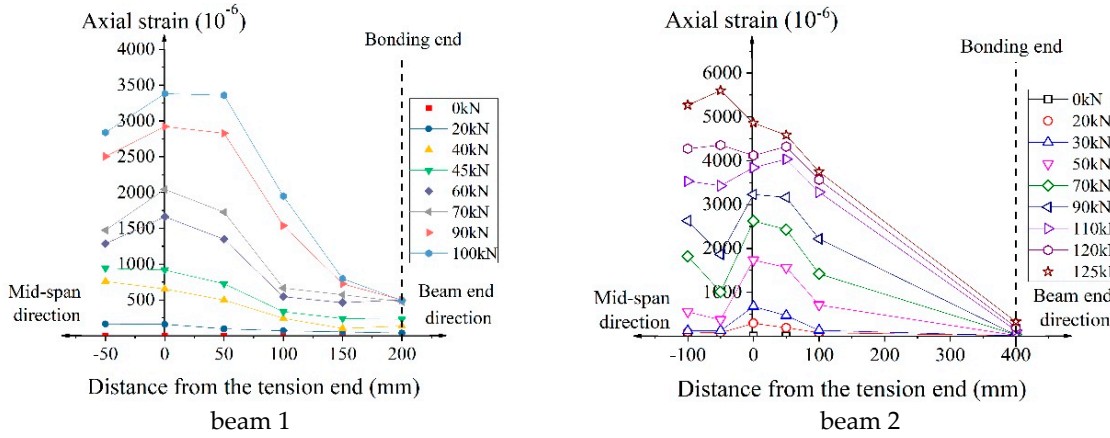

**Figure 11.** Carbon fiber reinforced polymer (CFRP) strain distribution of beam 1 and beam 2.

**Table 4.** Characteristic values for loads.

| Specimen Designation | Tension End Cracking Load (kN) | Increased Range (%) | Bonded End Cracking Load (kN) | Increased Range (%) | Ultimate Load (kN) | Increased Range (%) |
|---|---|---|---|---|---|---|
| Beam 1 | 20 | - | 45 | - | 105 | - |
| Beam 2 | 30 | 50 | 125 | 177.78 | 130 | 23.81 |

### 4.2. Time Domain and Frequency Domain Analysis

In order to reduce the space of this paper, the time domain and frequency domain signal analysis only gives signals for the three typical loading stages of the SA sensor. During the experiment, the three typical loading stages of the specimen are as follows—(1) When the specimen is not loaded, it is in a completely healthy state; (2) The specimen is loaded to the state where cracks begin to appear. (3) The specimen is loaded until the crack developed into failure state. The time domain signals from OS1 and OS2 of beam 1 are shown in Figure 12. The time domain signals from OS1 and OS2 of beam 2 are shown in Figure 13. Each figure reflects the sensor signal response of a period from the swept sine wave signal. The results show that the amplitudes of the signal by the sensors decreases with the increase of the applied load. The reason is that cracks occur at the OS1-4 region of the specimen with the increase of load. The crack causes the stress wave signal reflect and attenuate. The initial crack and failure state of the specimen can be found by time domain analysis. In this test, the SA actuators generated swept sine wave signal to the sensors and the received signal is highly sensitive to the debonding crack condition between the CFRP and concrete. Fourier transform is also used for further frequency analysis of received signals. Compared with time domain signals, the downward trend of power spectral density (PSD) energy can be seen more easily observed in frequency domain. The frequency domain signals received by OS1 and OS2 of beam 1 are shown in Figures 14 and 15, respectively. The frequency domain signals by OS1 and OS2 of beam 2 are shown in Figures 16 and 17, respectively. By comparing the amplitude of the signal in the frequency domain, it is clear that the amplitude of the signal decreases with the increase of the applied load. It can be seen from the time domain and frequency domain analysis that the tension end region cracks earlier than the bond end region with the increase of load. Stress concentration at the end of CFRP leads to debonding damage at the early loading stage. This indicates the starting point of structural damage. The tensioned end region and the bonded end region of the beam 1 are cracked earlier than the beam 2. It is shown that increasing the length of the non-pre-stressed CFRP bond can slow down the appearance and expansion of the crack.

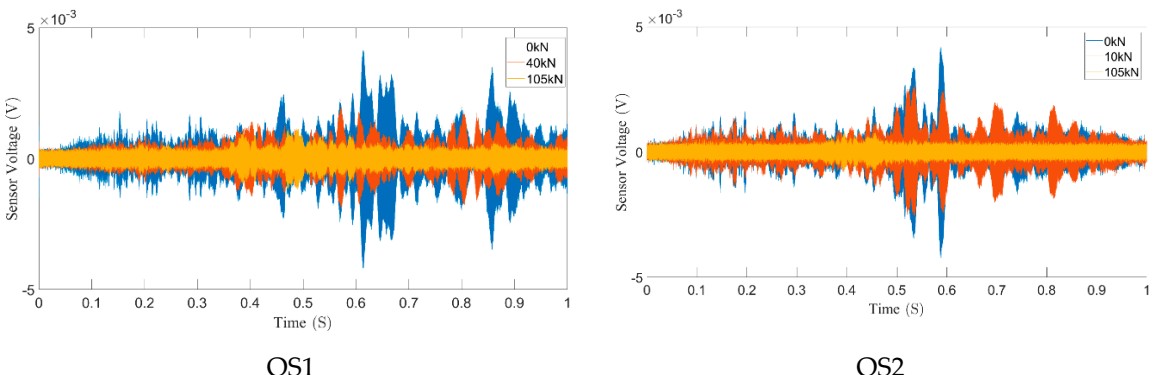

**Figure 12.** Time domain signals of OS1and OS2 in the beam 1.

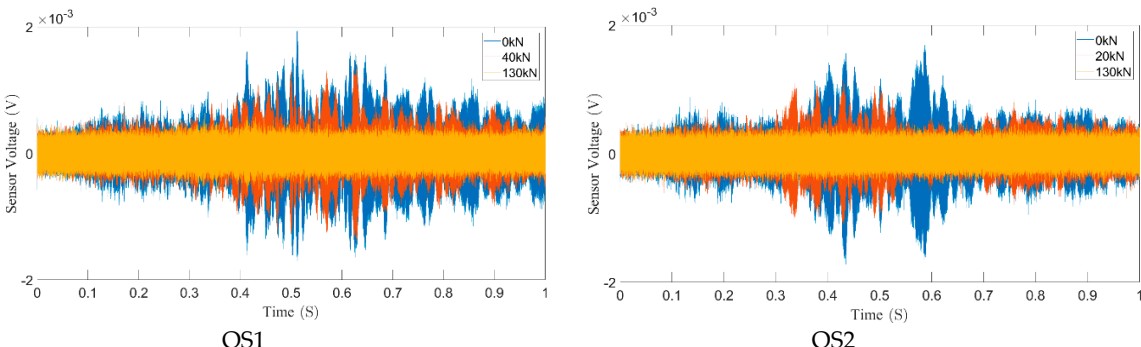

**Figure 13.** Time domain signals of OS1 and OS2 in the beam 2.

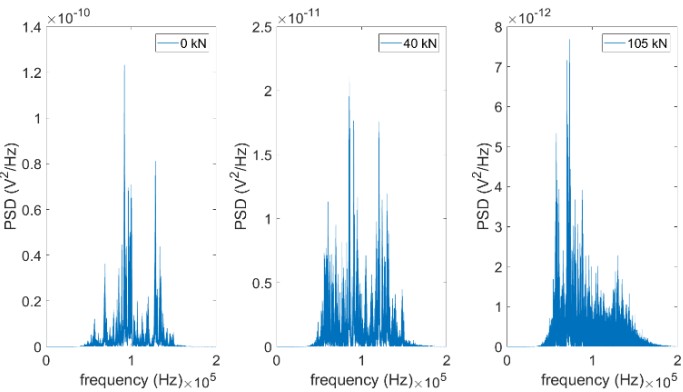

**Figure 14.** Frequency domain signal of OS1 in the beam 1.

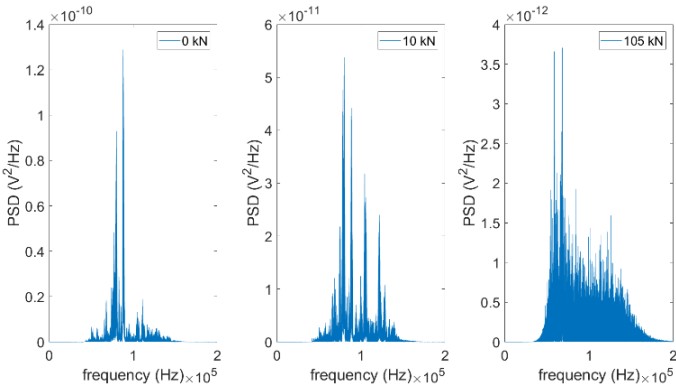

**Figure 15.** Frequency domain signal of OS2 in the beam 1.

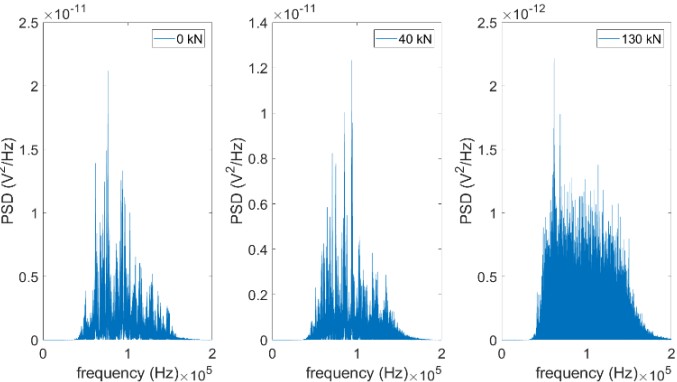

**Figure 16.** Frequency domain signal of OS1 in the beam 2.

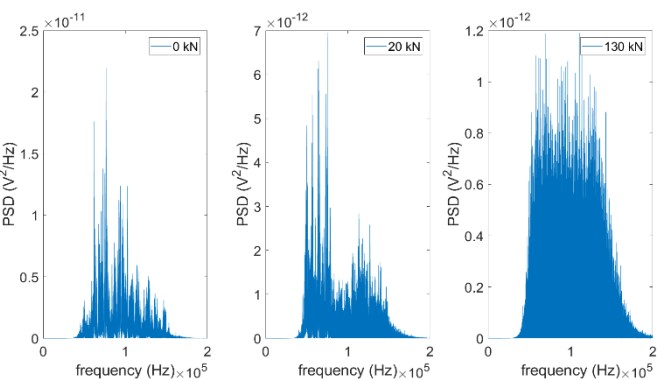

**Figure 17.** Frequency domain signal of OS2 in the beam 2.

### 4.3. Wavelet Packet-Based Damage Index

In order to quantify the degree of structural damage during the applied load process, the damage index is computed by using the wavelet packet analysis. The results are shown in Figures 18–21. It can be seen from Figures 18–21 that the value of the damage indices increase as the loading progressing. When the applied load reached 10 kN, the damage index of the OS1 and OS2 area sensors mounted on the beam 1 increased by 14.5% and 28.1%, respectively. The damage index of the OS1 and OS2 area sensors mounted on the beam 2 increased by 2.4% and 6.1%, respectively. This indicates that crack appears in the tensioned end region of the beam 1 when the applied load is only 10 kN. When the applied load reached 20 kN, the damage index of the OS1 and OS2 area sensors mounted on the beam 1 increased by 13.4% and 38.6%, respectively. The damage index of the OS1 and OS2 area sensors mounted on the beam 2 increased by 8.1% and 63.4%, respectively. This indicates that when the applied load is only 20 kN, cracks appear in the tensioned end region of the beam 2. A new crack continues to appear in the tensioned end region of the beam 1. As the applied load increases, the crack of the beam further expands. When the applied load reaches 40 kN, the damage index of the OS1 sensor installed in beam 1 and beam 2 increases by 36.4% and 46.5%. When the applied load is only 40 kN, cracks occur in the bonded end regions of the beam 1 and beam 2. By comparing the damage indices of the OS1 and OS2 area sensors, it can be seen that the damage index of OS2 grows faster with the increase of applied load. It is indicated that the tensioned end region cracks earlier than the bonded end region due to the influence of the stress change. The OS2 region damage index of beam 1 increases fastest under 10 kN and 20 kN loads. The OS2 region damage index of beam 2 increases fastest under 20 kN load. It is shown that increasing the length of the non-pre-stressed bond end has an effect on the cracking of the tensioned end region. The OS1 region damage index of beam 1 and beam 2 increases fastest at the applied load of 40 kN. The complete debonding failure of beam 1 was found at the applied load of 105 kN. The complete debonding failure of beam 2 was found at the applied load of 130 kN. It shows that increasing the non-pre-stressed bond length can improve the bearing capacity and the damage degree.

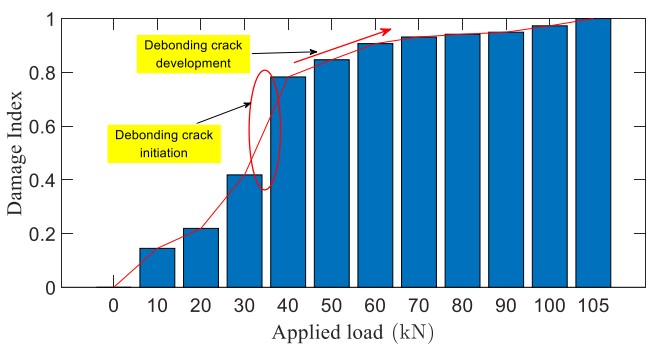

**Figure 18.** Damage index of OS1 in the beam 1.

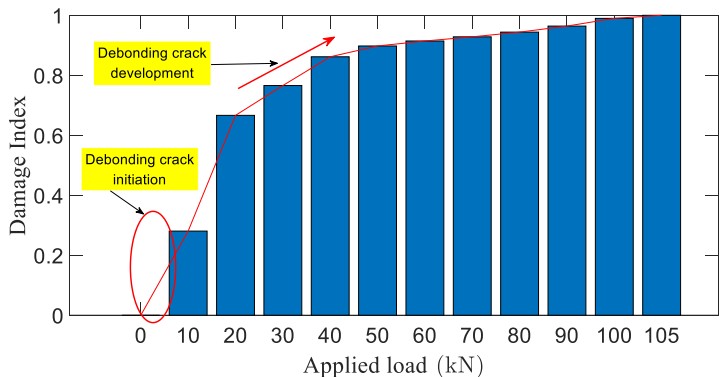

**Figure 19.** Damage index of OS2 in the beam 1.

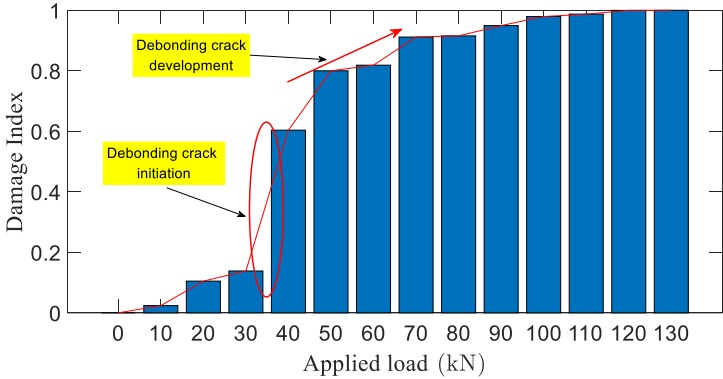

**Figure 20.** Damage index of OS1 in the beam 2.

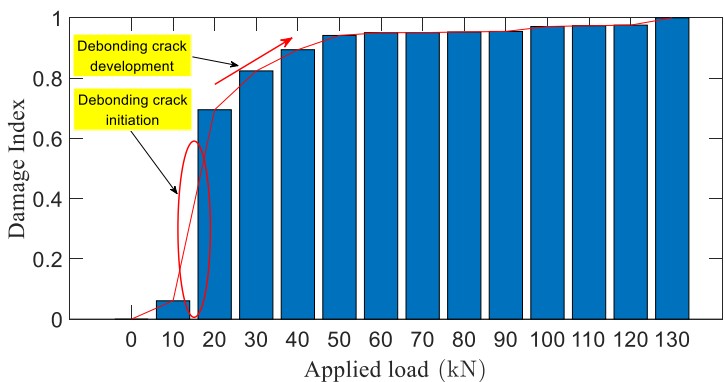

**Figure 21.** Damage index of OS2 in the beam 2.

## 5. Conclusions

This paper presents an experimental investigation of detecting crack-induced damage using active sensing approach with embedded SAs through two RC beams with NSM pre-stressed CFRP plates. Actuators and sensors for damage detection can be formed by embedding SAs in the test specimens. The embedded SA-induced stress wave signal is highly sensitive to the debonding crack condition between the CFRP and concrete. Based on the experimental results, the following conclusions can be drawn:

1.  The debonding cracks only appear in the CFRP reinforcement region. The non-pre-stressed bonding section can prolong the bond stress-transmission length of the CFRP-concrete interface. The increase in the contact area between the CFRP and the concrete improves the stress distribution of the concrete. The analysis results show that there is an effective non-pre-stressed CFRP bond

length (300 mm), which can improve the load characteristics of the test specimen and enhance the bearing capacity and deformation performance of the test specimen.

2. Wavelet packet theory has been successfully applied to the real time monitoring of the debonding damage process of the RC beams with NSM pre-stressed CFRP plates. It can be seen from the time domain and frequency domain analysis that the tension end region cracks earlier than the bond end region with the increase of load. Stress concentration at the end of CFRP leads to debonding damage at the early loading stage. This indicates the starting point of structural damage. It is shown that increasing the length of the non-pre-stressed CFRP bond can slow down the appearance and expansion of the crack. It can be judged from the damage index that increasing the contact area between CFRP and concrete affects the damage degree and bearing capacity of the non-pre-stressed CFRP bonding section; the existence of the non-pre-stressed CFRP bonding section affects the damage degree and bearing capacity of the tension end and the bond end region. Wavelet packet damage index has the ability to detect debonding crack initiation. After the occurrence of debonding cracks, the value of damage index increasing continuously monitors the development of debonding cracks.

3. The active sensing approach based on SAs can monitor the occurrence and development of cracks for RC beams with NSM pre-stressed CFRP plates in real time. The experimental results show that it is feasible to use the active sensing method based on stress wave to monitor the debonding crack damage of the CFRP-concrete in real time.

In addition, the effectiveness of the proposed active sensing approach was experimentally verified by the debonding performance of different lengths of non-pre-stressed CFRP bonded. The method is of significance to monitor the initial installation quality and the long-term efficiency of the CFRP-concrete. The predicted structural damage occurred earlier than the real damage. Meantime, the piezoceramic SAs are economical with low manufacturing cost and they can be easily integrated with concrete structures. The low sampling rate requirement for the data analysis reduces the cost of the data acquisition system. It has the potential to be applied to the health monitoring at a very economical cost without using additional bulky equipment. Therefore, the proposed active sensing approach based on SAs has great potential to be applied in practice for inaccessible damage detection of RC beams with NSM pre-stressed CFRP plates.

## 6. Future Work

➢ The long-term performance of RC beams with NSM pre-stressed CFRP plates, including the aging and durability of CFRP and adhesives, needs further study.

➢ Effective and high-strength bonding and anchoring measures are studied to make the strength of CFRP material fully play.

➢ Furthermore, the shrinkage and creep of concrete and adhesive, the creep of CFRP material and the influence of temperature characteristics on the creep performance of reinforced concrete beams were studied.

➢ The overall durability of concrete beams strengthened with CFRP is a general concern. Due to the differences in physical properties, mechanical properties and other properties between the existing concrete beam material and the new reinforcement material, the durability evaluation of the existing concrete beam after reinforcement is not only different from the durability evaluation of the existing concrete beam but also different from the durability design of the new concrete beam, which makes the prediction of the durability of the reinforced concrete beam more complicated.

➢ In order to provide a feasible design basis for the fatigue design of reinforced concrete beams, it is necessary to carry out a systematic study on the fatigue and other dynamic properties of concrete beams strengthened with CFRP.

➢ In the case of a considerable number of experimental studies, it is necessary to use the numerical analysis method to carry out the numerical experimental research on the concrete beams

strengthened by CFRP materials and to carry out the numerical simulation research on different combinations of concrete strength grade, reinforcement materials, mechanical properties, slotting size, clear distance between slots and so forth, so as to seek a reasonable and practical theoretical analysis method and design calculation formula.

➢ On the basis of accurate structural analysis, it is absolutely necessary to study the reliability of reinforced structure, so that the two complement each other and the structural design method is perfected.

**Author Contributions:** Y.L. designed and performed the experiments and helped write the manuscript. M.Z. helped collect the data, analyzed the data and wrote most part of the paper. X.Y. developed the concept and made critical revision of the manuscript. Z.H. analyzed the data, assisted with plots. L.W. helped design and perform the experiments and helped analyze the data. All authors have read and agreed to the published version of the manuscript.

**Funding:** The authors are grateful for the partial financial support received from the Major State Basic Research Development Program of China (973 Program, grant number 2015CB057704), the National Nature Science Foundation of China [Grant No. 51378081], the Natural Science Foundation of Hunan Province (Grant No. 2019JJ40313), the Hunan Provincial Innovation Foundation for Postgraduates (CX20190651).

**Conflicts of Interest:** The authors declare no conflict of interest.

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
