# Peer review of "Debonding Detection of Reinforced Concrete (RC) Beam with Near-Surface Mounted (NSM) Pre-stressed Carbon Fiber Reinforced Polymer (CFRP) Plates Using Embedded Piezoceramic Smart Aggregates (SAs)"

_applsci, doi:10.3390/app10010050_

Round 1

Reviewer 1 Report

In the Reviewer opinion the research paper entitled “Debonding Detection of Reinforced Concrete (RC) Beam with Near-Surface Mounted (NSM) Prestressed CFRP Plates Using Embedded Piezoceramic Smart Aggregates” is good.

This paper presents an experimental investigation of detecting crack-induced damage using active sensing approach with embedded SAs through two RC beams with NSM prestressed CFRP plates. Actuators and sensors for damage detection can be formed by embedding SAs in the test specimens. The embedded SA-induced stress wave signal is highly sensitive to the debonding crack condition between the CFRP and concrete. The article treats some aspects of a very interesting and emerging topic in the civil engineering constructions. The conducted tests are correct and there are concisely explained in the text.

Some comments which greatly enhance the understanding of the paper and its value are presented below. Specific issues that require further consideration are:

The title of the manuscript is well matched to its content. The structure of the manuscript is proper. The Introduction sufficiently covers the cases. In the Reviewer’s opinion, the current state of knowledge relating to the manuscript topic has been covered and clearly presented, but the author's contribution and novelty are not emphasized. In this regard, the authors should make their effort to address this issue, by adding additional comments on the state of the art and the proposed aspects. An analysis of the manuscript content and the References shows that the manuscript under review constitutes a summary of the Author(s) achievements in the field. Fig. 2 – I don’t understand this drawing. Please explain what doesn’t means? Fig. 7 – all element are described, but is hard to read (left picture). Fig. 17, 18 – please improve the quality of fig., what means red line? In the Reviewer’s opinion, the bibliography, comprising 25 references, is representative and exhaustive, constituting a good source of information, but almost all of them comes only from Asia. Additional comments should be added in regard to the practical value of this research, how the industry can profit from that. In the Reviewer’s opinion the manuscript should be published in the journal after minor revision.

Reviewer 2 Report

This is a very well written manuscript about the experimental test on RC members strengthened by carbon FRPs. The design of experiment is clear, and contains valuable information. 

I noticed that the location of FRPs are fixed. This is probably the best potential location (not sure if it is proved in paper!? or even maybe the experiments are expensive); however, the location of FRP jackets is very important factor towards a resilient design. I recommend that the authors discuss and justify the current choice and provide some recommendations for future work. There might be useful information in: Optimal FRP Jacket Placement in RC Frame
Structures Towards a Resilient Seismic Design
.

Also, it is useful if the authors provide some hints for future numerical simulations (both in micro and macro level) for FRP added RC members.
